# Influence of Bladder Filling on Parameters of Body Composition by Bioimpedance Electrical Analysis: Observational Study

**DOI:** 10.3390/s24227343

**Published:** 2024-11-18

**Authors:** Asunción Ferri-Morales, Sara Ando-Lafuente, Cristina Lirio-Romero, Emanuele Marzetti, Elisabeth Bravo-Esteban

**Affiliations:** 1Faculty of Nursing and Physiotherapy, University of Castilla La Mancha, 45071 Toledo, Spain; asuncion.ferri@uclm.es (A.F.-M.); sara.ando@uclm.es (S.A.-L.); elisabeth.bravo@uclm.es (E.B.-E.); 2Health and Social Research Center, University of Castilla La Mancha, 16071 Cuenca, Spain; 3ImproveLab, Research Group of Pediatric and Neurologic Physiotherapy, 45071 Toledo, Spain; 4Department of Geriatrics, Orthopedics and Rheumatology, Fondazione Policlinico Universitario “A. Gemelli” IRCCS, Università Cattolica del Sacro Cuore, 00168 Rome, Italy; emanuele.marzetti@unicatt.it; 5Physiotherapy Research Group, Toledo (GIFTO), University of Castilla La Mancha, 45071 Toledo, Spain; 6Institute of Health Research of Castilla-La Mancha (IDISCAM), 45071 Toledo, Spain

**Keywords:** body composition, bladder, physical fitness, skeletal muscle, body mass index

## Abstract

Bioelectrical impedance analysis (BIA) is a widely used method for estimating body composition, and its accuracy may be influenced by various factors, including bladder filling. This study aims to investigate the impact of bladder filling on the accuracy of BIA measurements. An experimental crossover study was conducted with sedentary young adults. The influence of bladder filling on total body water (TBW), fat mass (FM), fat-free mass (FFM), and basal metabolic rate (BMR) was assessed. Participant in underwear followed an overnight fast. They were instructed to abstain from vigorous physical activity and alcohol for at least 24 h prior to the session. The results obtained from single-frequency and multi-frequency BIA devices were compared. The findings suggest that bladder filling does not affect measured impedance; however, changes in weight following bladder voiding influenced derived BIA results. Specifically, TBW, FM, and BMR values significantly reduced after voiding (*p* < 0.05). Furthermore, the study found poor agreement between single-frequency and multi-frequency BIA devices, indicating that they are not interchangeable. Bladder filling does affect BIA measurements, not clinically meaningful. Further research is needed to explore the implications of these findings for clinical practice and research protocols.

## 1. Introduction

The human body is composed of fat mass (FM) and fat free mass (FFM) which consists of all that is not fat (bone mineral content, body cell mass, and total body water (TBW) which is composed of extracellular and intracellular water. Bioelectrical impedance analysis (BIA) is frequently used for assessing body composition and nutritional status [1,2], in healthy subjects and clinical conditions (diabetes, obesity, renal failure, sarcopenia) [3,4,5,6], by applying specific equations for a given age, weight, height, sex, type of population [7]. It is a non-invasive, safe and relatively simple technology, in which an alternate current of less than 1 mA passes across the body [8], and quantifies the biological impedance, i.e., the resistance of biological tissues to the flow of the electric current [9,10]. 

BIA devices are calibrated to estimate, after obtaining impedance values, the distribution of body fluids and TBW [7,8]. FFM can be estimated, assuming that TBW is a constant part of FFM in normal hydration subjects (TBW/FFM = 0.73) [1,8]. Then, FM can be obtained by subtracting FFM from total body weight.

The two common BIA techniques differ in the frequency of the current used: single or multi-frequency. Single-frequency bioelectrical impedance analysis (BIA_SF_) is the most used, usually employs a 50 kHz current and have been used to assess TBW and FFM using equations [8,11]. This current that does not penetrate well the cell membranes, then shows limitations in the intracellular water prediction [7]. In contrast, multifrequency bioelectrical impedance analysis (BIA_MF_) uses current of different frequencies that penetrates cell membranes and flows through all fluids in intra-and extracellular spaces, then it can be used to estimate FFM, TBW, intra and extra-cellular water [11,12,13]. 

Different factors may influence the measurements obtained by BIA, such as altered hydration status and amount of fat mass [7,14]. Considering that current will pass predominantly through tissues with higher conductibility [15], any small modification within the hydration level of these tissues or any compartments rich in water, may probably cause significant changes in the value of impedance [16]. Thus, variables related to hydration level, the temperature of the skin [17], food and fluid intake [18], intense physical activity before the evaluation, or the body position during the measurement, may result determinant for estimating the different components of the body [10,19,20], and can be reflected within the guidelines provided to the participants of the research studies, such as going to the fasting test or emptying the bladder before performing the tests [19,20]. 

Studies focused on the determination of the role that food intake has on the validity of BIA systems have been carried out [21]. However, there is a lack of studies that investigate whether the parameters of body composition are modified based on bladder filling. In addition, it has been showed that impedance values and therefore, body composition parameters, can vary depending on current frequency flowing through the organism or the type or BIA used [8,9,22]. Thus, the aim of this study was to evaluate whether bladder filling influences on the parameters of body composition estimated by BIA, and to compare the parameters of body compositions obtained by two bioimpedance analyzers of different frequency (single-frequency and multi-frequency), and also, to assess the agreement between both devices regarding body composition estimates. 

## 2. Methods

### 2.1. Study Design

This is an experimental crossover study that examined the effects of bladder filling on body composition in apparently healthy sedentary young adults. The study protocol was approved by the Research Ethics Committee of the University of Castilla-La Mancha (UCLM), Spain under the protocol number 132/2017. All study procedures were conducted in compliance with the Declaration of Helsinki and Resolution 196/96 of the National Health Council.

### 2.2. Study Participants

A convenience sample of seventy volunteers participated in this study. This sample size is in accordance with the recommended size for reliability and agreement studies [23]. Participants were recruited via advertisements and direct contact. People eligible to participate of the present study were sedentary university students, identified according to the American College of Sport Medicine from the Faculty of Physiotherapy and Nursing of UCLM in Toledo with ages ranging from18 to 22 years old. All the participants were informed about the study and signed the informed consent.

People who were engaged in more than two exercise sessions per week, reported the use of anticholinergic drugs, were unable to stand up, pregnant, and presented a clinical diagnosis of neurological diseases (neurogenic bladder) were excluded. We also excluded participants with a voiding volume lower than 250 mL and those whose reports were inaccurate. 

### 2.3. Experimental Procedures

The experimental session (Figure 1) occurred in a single day in underwear. Participants reported to the laboratory in the morning (8–10 a.m.) after an overnight fast. They were also instructed to avoid vigorous physical activities and alcohol intake for a minimum of 24 h before. Following height evaluation and one hour before body composition assessment, participants consumed 500 mL of mineral water to fill the bladder. Transabdominal ultrasound was performed to ensure that the bladder was fully filled. Then, BIA analysis was performed while the bladder was filled. A second analysis was conducted after participants voiding the bladder in a volumetric container. The analysis was valid if micturition volume was equivalent to the amount recorded in ultrasonography analysis. 

### 2.4. Anthropometric Measurements

Height (m) was measured using a wall stadiometer with a precision of 0.1 mm (SECA GmbH & Co. KG., Hamburg, Germany) and weight (kg) was measured with the BIA taken before and after bladder emptying, at the same time as the two analyses performed for each subject. Body mass index was automatically calculated by the BIA according to the following formula:BIA = weight (kg)/(height [m])^2^

### 2.5. Evaluation of Bladder Volume Using Ultrasonography Analysis

Transabdominal ultrasound evaluations were performed using a portable Ultrasound (GE Medical Systems, Chicago, IL, USA) in a B mode with a 3–5 MHz convex array transducer by an experienced assessor. Assessments were conducted with participants lying supine with the suprapubic area exposed, hip flexed approximately at 60 degrees, and feet on the surface of the table. Measurements involved two 2D images of the bladder [24] performed in the sagittal, in the midline, and axial (transverse) planes at suprapubic level. Bladder volume was determined automatically by the ultrasound software using an ellipsoid formula [24]:Bladder volume (mL) = height × depth × transverse × 0.5. 

### 2.6. Bioelectrical Impedance Analysis 

Body composition was estimated using both BIA_SF_ (50 KHz, 500 μA, Tanita^®^ BC-418 bioimpedance analyzer, Tanita Corp. Tokyo, Japan) and BIA_MF_ (20 kHz and 100 kHz, 330 μA, Inbody^®^ 230 bioimpedance analyser, InBody Co., Ltd., Seoul, Republic of Korea), using eight-polar equipment. Measurements were taken under standard conditions in a temperature-controlled room (24.0 °C), with participants wearing light clothes or a swimsuit. For the analysis, participants remained standing, with barefeet touching four electrodes of the BIA platform, elbows flexed at 45°, and hands grabbing the other four electrodes [9]. BIA outputs include weight, TBW, FFM, FT, and basal metabolic rate (BMR).

### 2.7. Statistical Analysis

Data are shown as mean ± standard deviation (SD) or median ± Inter-Quartile Range (IQR). The normality of data was tested using the Kolmogorov-Smirnov test and graphical analysis. All variables followed a non-normal distribution, except BIA impedance. The measurements from different devices, depending on the normal data distribution, were compared using the Student’s *t*-tests (two-sided) or non-parametric tests (Wilcoxon test, Mann-Withney U test), and Spearman or Perarson’s correlation to examine the relationship between variables. 

Intraclass correlation coefficients (ICC) [25,26] was used to assess the degree of agreement, before and after bladder voiding. ICCs ≥ 0.8 indicated a strong level of reliability. 95% limits of agreement (LOA), calculated as mean bias ± 1.96 was used to assess the range of agreement, within which 95% of the differences between two measurement obtained by BIASF and BIAMF are included. The level of significance was set at alpha = 5% (*p* < 0.05) and all analyses were performed using SPSS IBM (software, v.23.0 SPSS Inc., Chicago, IL, USA). Bland-Altman analysis [27] was used as a complementary analysis for clinically meaningful differences were calculated using the MedCalc software 12.3 (Ostend, Belgium). 

## 3. Results

Seventy-one young adults (20.4 ± 3.8 years), 40 women (56.3%), participated of the present study. Data of two participants, one in the full bladder and one in the voiding bladder, were excluded from the BIA_MF_ analysis due to measurement error. Table 1 shows descriptive values and normality result of anthropometric measurements. Results of the correlation analysis are shown in Table 2. Significant correlations were observed between TBW, FFM and BMR, regardless of BIA frequency. However, significantly and negatively correlations between TBW and FM, and positive associations between FFM and FM, and BMR and FM were only observed in data acquired using the BIA_MF_ method. FFM presented the same pattern of relationship with FM and BMR as the TBW. Pre- and post-intervention BIA values are shown in Table 3.

TBW, FM, and BMR were significantly reduced after bladder voiding, regardless of the BIA frequency. However, specific results were noted for FFM, with significant reductions observed according to BIA_SF_ method, whereas a significant increase was indicated by the BIA_MF_ equipment. 

Significant differences were found in TBW, FFM, FM and BMR when comparing full and empty bladder, with higher values before voiding in all variables (*p* < 0.05, Table 3) The Impedance values shown in Table 3 correspond to the whole body in the BIA_SF_, and to the trunk in the BIA_MF_.

Results of Bland-Altman analysis are shown in Table 3. Results indicated that LOA were slightly wider for all variables registered with BIA_MF_ than with BIA_SF_ (Table 2), suggesting a lower accuracy of the BIA_MF_ method. An ICC ≥ 0.94 was observed for all variables. 

Results of Bland-Altman analysis and plots for comparisons between methods are shown in Table 2 and Figure 2. A significant mean bias was found when comparing TBW, FM, FFM and BMR obtained by BIA_SF_ versus BIA_MF_ with higher values for TBW, FFM an BMR measured with BIA_SF_ than with BIA_MF_. The agreement between BIA_SF_ and BIA_MF_ was not clinically acceptable neither for FM and for FFM. The LOAs are wide for each of the variables in both conditions and were slightly greater for the FM than for the FFM measurements (In Set 1, BIA_SF_ versus BIA_MF_: FM LOA = −5.5 to 2.6 kg and FFM LOA = −2.4 to 5.5 kg. In Set 2, FM LOA = −5.7 to 2.3 kg and FFM LOA = −2.1 to 5.8 kg, Table 3). In this study, the mean FM was somewhat more than a third of the FFM (FFM = 50.7 kg, FM = 14.3 kg in BIA_SF_; and FFM = 49.1 kg, FM = 15.8 kg, Table 4) and thus, the degree of error was larger in FM estimation by BIA_SF_ and BIA_MF_ compared with that in FFM. No trend was found in the difference between methods as the average increases.

## 4. Discussion

The main findings of the present study indicate that bladder filling impacts BIA results. Specifically, we found that TBW, FM, and BMR values were significantly reduced after bladder voiding. However, changes in FM seems to be dependent on the type of equipment, with significant increases observed using BIA_SF_, whereas a significant decrease was found when the BIA_MF_ was utilized. Moreover, when the accuracy to assess body composition of both methods were compared, meaningful disagreements were observed. 

### 4.1. Full Bladder Versus Empty Bladder

TBW estimation using bioimpedance measurements are based on the inversely proportional measurement between body resistance and the total amount of body water and as it was expected, in the current study, it was found inverse associations between impedance and the variables with higher water content, such as TBW and FFM, in both sets and with both analyzers; which confirms that there is a lower resistance to the current flow in regions with larger quantity of fluid in the organisms, meaning higher conductivity in water-rich tissues and electrolytes, such as muscle tissue, as other authors confirmed [7,28]. However, the present study failed to find statistical differences in impedance variable (Ω) (Impedance of whole body in BIA_SF_ and Impedance of the trunk in BIA_MF_) were found between full and empty bladder. This finding may indicate that the findings are caused by the difference in weights rather than impedance. Regarding technical considerations of the BIA methodology and analysis conditions, we note that BIA devices use predictive equations that may factor in minor fluid changes without altering impedance values substantially. Additionally, these findings suggest that bladder filling could interact with the assumptions within these equations [7], impacting TBW estimations, even when impedance itself remains constant [29]. Apparently, the amount of urine accumulated in the bladder was not enough to observe these changes. Kyle et al. indicated that total bioimpedance measurement assesses mainly the upper and lower limb compartments and shows some limitations in predicting water compartments of the trunk [7], where the bladder is. Pirlich et al. in cirrhotic patients with ascites showed that even marked changes in fluid volume inside the abdominal cavity have only a minor influence on the measurement of FFM [30]. 

Despite not having observed differences in the impedance between both sets, both BIAS are sensitive enough to detect more TBW with full and with empty bladder. BIA_MF_ detected higher difference in TBW and in FFM; as well as minor change in FM between full and empty bladder than BIA_SF_. Several authors established that, in general, the BIA_MF_ method predicts extracellular fluids more precisely than the BIA_SF_ method [8]. 

These findings confirm that the amount of FFM and FM obtained with full bladder are higher than those obtained with empty bladder in both analyzers, with significant differences between the measurements recorded with full and with empty bladder. In both BIAs, FM presents higher variability than FFM and a wider LOA between which 95% of the differences would be expected. 

An important issue emerging from the NIH Technology Assessment Conference is that subject measurement conditions must be rigorously standardized to obtain accurate body composition estimates [9]. Nowadays, it is widely probed that different factors may influence the measurements obtained by BIA, such as liquid or food consumption [18], heat exposure [17], physical activity [14,31] or body position [19]. This is the first study to verify the influence of bladder filling on the parameters of body composition measured by bioimpedance. Although changes in the bladder volume failed to reflect major differences in the body composition parameters, in order to minimize biases and increase the precision of the measurement, it is convenient to request the subjects to empty the bladder before starting the bioimpedance test. 

In this study, it has also been controlled the basal metabolic rate. In this context, results showed a strong association between this variable and FFM. This suggests that the amount of muscle is directly proportional to the metabolic rate; and as with the FFM, the analyzer detects higher BMR with full than with empty bladder. Although differences were not large, the fact that they were detected indicates a good sensitivity of both analyzers to identify the change. These findings are consistent with previous studies reporting that FFM is the main determinant of BMR [29,30]. Johnstone et al. [32] suggest that each kilogram of lean tissue exerts about 5 times more effect on BMR than each kilogram of fat tissue.

Diuresis might increase resistance in ionic tissues by reducing the amount of fluid available to conduct the electric current, consequently promoting changes in body composition according to BIA analysis [33,34]. Although in the present study significant decreases were observed in all parameters (Table 3), previous research suggests that variations within approximately 1–2% of body composition metrics are typically within the measurement error of most BIA devices and are unlikely to be clinically meaningful [7,35]. Since the observed differences values fell within these error margins, we interpreted them as not clinically significant, suggesting that bladder filling status does not substantially impact body composition estimates. However, in the present study, significant decreases were observed in FM. These findings might suggest that BIA methods are not sufficiently accurate to assess changes in TBW [7] and that changes found in the present study are not clinically significant. Indeed, BIA might not be sensible to represent changes in intra- and extracellular water are acutely altered [36]. 

### 4.2. Agreement Between BIA Single Frequency with BIA Multifrecuency

The FFM, TBW and FM estimates showed excellent relative reliability between BIA_MF_ and BIA_SF_ by ICC but clinically important differences by Bland-Altman plots. This may because high ICC does not necessarily mean agreement between methods [37]. Reliability assessed by ICC is a scaled agreement index depending on the range of the data (between-subject variability) and may produce high values for heterogeneous populations [38]. The large range of the measures in our subjects could produce a relatively high ICC value. In contrast, bias and LOAs (Bland-Altman plot) are unscaled indices based on the original unit and if the absolute limit is less than an acceptable difference, then the agreement between the two observers is deemed satisfactory [38]. 

LOAs estimate the interval in which 95% of the differences between the two BIAs are likely to be found. We observed that in the parameters of corporal composition a subject measured with BIA_SF_ and full bladder, can obtain, in 95% of the cases, a value of FFM in a range between 2.4 kg below to 5.5 kg above, that measuring with BIA_MF_ and similar differences with empty bladder. In the FM values the opposite will occur, with full bladder to subject can obtain with BIA_SF_ a difference with respect to the BIA_MF_ of 5.5 kg below to 2.6 kg above. The authors consider that the discrepancy is enough large to be important. Then the two methods are not interchangeable. In our study we have not compared with the gold standard method, we have compared two BIAs, commonly used in the analysis of body composition. In line with our results, A validation study of BIA Tanita^®^ BC-418 and BIA Inbody^®^230 [39] using DXA as gold standard also found, high coefficients of correlation and poor agreement (wide LOAs) in the estimation of FM between both BIAs and DXA in children. The LOAs were larger and the biases were greater for BIA_SF_ measurements compared with BIA_MF_. 

In clinical practice, the choice between the BIA_MF_ and BIA_SF_ devices should consider their distinct capabilities: while the BIA_MF_’s multi-frequency analysis and segmental assessment provide detailed insights into fluid compartments and localized body composition—beneficial in settings such as renal care, cardiology, or sports medicine—the BIASF device offers a faster, single-frequency measurement suited for general health assessments where detailed fluid differentiation is less critical [12].

Although FFM and FM values typically differ between men and women—males generally having less fat mass and more fat-free mass across all age ranges [40]—we conducted our analysis with both genders combined. This approach was chosen because our primary objective was not to examine FFM or FM quantities by gender but rather to determine if these values changed after bladder emptying, regardless of gender.

The present study has some limitations that should be acknowledge. First, BIA equipment is based on different formulas. Second, sudoresis and body temperature were not recorded. Third, diet habits were not collected. Lastly, participant position (supine vs. standing) can impact BIA measurements due to gravitational shifts in fluid distribution. The devices used in the present study are limited to measuring participants in a standing position, which restricts our ability to explore how gravity may impact fluid distribution and bioimpedance results in a supine position.

## 5. Conclusions

The measurements of TBW, FM, FFM, BMR shown higher values with full bladder than with empty bladder, but these differences are not clinically relevant. Both BIA methods demonstrated poor agreement, indicating they are not interchangeable. The BIA_SF_ provides consistently higher values for TBW and FFM, and consequently lower values for FM than the BIA_MF_.

## Figures and Tables

**Figure 1 sensors-24-07343-f001:**
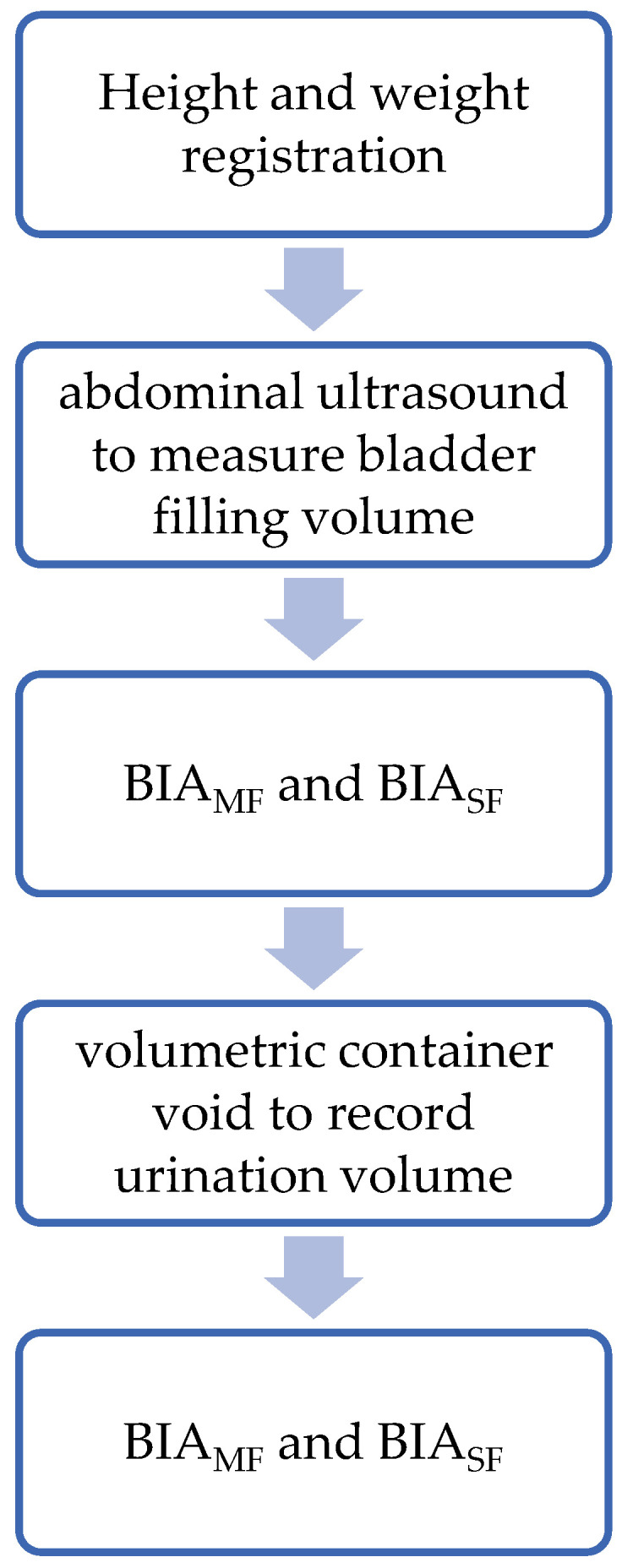
Experimental protocol. BIA_SF_: Single-frequency bioelectrical impedance analysis; BIA_MF_: Multifrequency bioelectrical impedance analysis.

**Figure 2 sensors-24-07343-f002:**
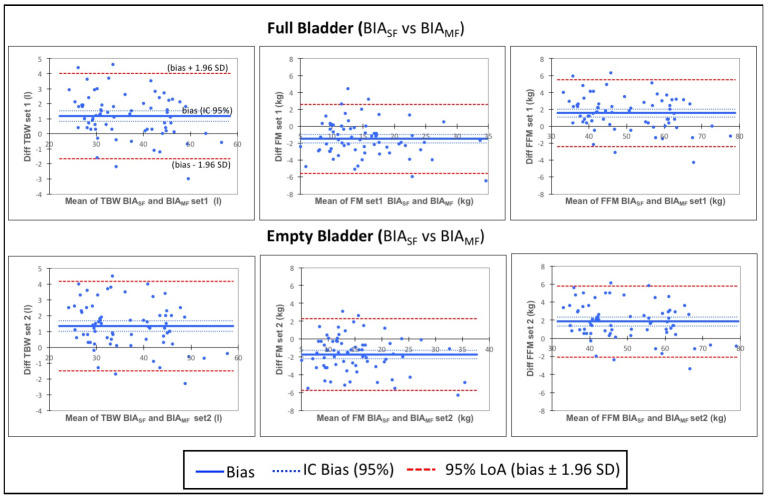
Bland Altman plots of BIA_SF_ versus BIA_MF_ estimates of body composition with full and empty bladder.

**Table 1 sensors-24-07343-t001:** Study population characteristics. Descriptive values and normality result of anthropometric measurements.

Variable		All Participants (N = 70)			Women (N = 40)		Men (N = 30)		
		Set 1 Median (SD)	Set 2 Median (SD)	*p*	Set 1 Median (SD)	Set 2 Median (SD)	Set 1 Median (SD)	Set 2 Median (SD)	*p*
	Age (y)	20.43 (3.85)	-		20.30 (4.64)	-	20.6 (2.49)	-	0.06
	Height (m)	1.67 (0.09)	-		1.61 (0.56)	-	1.75 (0.07)	-	0.13
	WHR	0.86 (0.04)	0.85 (0.05)	0.12	0.85 (0.05)	0.84 (0.05)	0.87 (0.04)	0.86 (0.04)	0.01
	Volume (mL)	449 (276)	546 (208)	<0.001					
BIA_SF_	Weight (kg)	65.18 (11.40)	64.59 (11.33)	0.001	59.2 (89.69)	58.66 (9.67)	73.16 (8.25)	72.51 (8.13)	0.32
	BMI (kg/m^2^)	23.27 (3.15)	23.07 (3.17)	0.001	22.79 (3.56)	22.58 (3.56)	23.91 (2.39)	23.73 (2.46)	0.04
BIA_MF_	Weight (kg)	65.13 (11.36)	65.51 (11.35)	0.001	59.17 (9.70)	72.49 (8)	73.09 (8.12)	58.52 (9.72)	0.28
	BMI (kg/m^2^)	23.26 (3.18)	23.03 (3.16)	0.001	22.77 (3.62)	22.54 (3.6)	23.91 (2.36)	23.68 (2.37)	0.04

BIA_SF_ (BIA single frequency, TANITA); BIA_MF_ (BIA multifrequency, INBODY); WHR: Waist-hip ratio estimated using Inbody software; BMI: Body mass index; SD: Standard deviation; Set 1: data collected pre-miction; Set 2: data collected post-miction; *p*: *p*-value of independent samples at baseline (Mann-Withney U test).

**Table 2 sensors-24-07343-t002:** Relationship between variables (bivariate correlations).

BIA_SF_										BIA_MF_							
		TBW		FM		FFM		BMR		TBW		FM		FFM		BMR	
		Set 1	Set 2	Set 1	Set 2	Set 1	Set 2	Set 1	Set 2	Set 1	Set 2	Set 1	Set 2	Set 1	Set 2	Set 1	Set 2
Z	r	−0.86	−0.87	0.11	0.11	−0.86	−0.87	−0.85	−0.86	−0.76	−0.74	0.12	0.03	−0.76	−0.74	−0.74	−0.74
	*p*	<0.001	<0.001	0.36	0.35	<0.001	<0.001	<0.001	<0.001	<0.001	<0.001	0.314	0.83	<0.001	<0.001	<0.001	<0.001
TBW	r_s_			−0.18	−0.18	1	1	1	1			−0.29	−0.32	1	1	1	1
	*p*			0.142	0.125	<0.001	<0.001	<0.001	<0.001			0.02	0.007	<0.001	<0.001	<0.001	<0.001
FM	r_s_					−0.18	−0.18	−0.14	−0.15					−0.29	−0.32	−0.29	−0.32
	*p*					0.142	0.126	0.256	0.224					0.2	0.007	0.2	0.008
FFM	r_s_							1	1							1	1
	*p*							<0.001	<0.001							<0.001	<0.001

Z, Impedance of whole body in BIA_SF_ (BIA single frequency, electric current of 50 kHz) and impedance of trunk in BIA_MF_ (BIA multifrequency, electric current 20 kHz and 100 kHz); TBW: total body water; FM: body fat mass; FFM: fat-free-mass; BMR: basal metabolic rate; r, Pearson’s correlation coefficient; r_s_, Spearman’s rank correlation coefficient. Set 1: dates pre miction; Set 2: dates post voiding.

**Table 3 sensors-24-07343-t003:** Median comparison and Bland Altman analysis between set1 and set2 in BIA_SF_ and BIA_MF_.

BIA_SF_ (*n* = 71)	Set 1	Set 2		Bland-Altman Analyse				
Variable	Median (IQR)	Median (IQR)	*p*	Lower LOA	IC (95%)	Upper LOA	IC (95%)	Range
TBW (L)	34.6 (30.1–44.7)	34.4 (30–44.4)	0.005	−0.60	(−0.75 to −0.46)	0.84	(0.69 to 0.98)	1.44
FM (kg)	13 (10.1–17.5)	12.9 (10–16.8)	<0.001	−0.69	(−0.92 to −0.46)	1.58	(1.35 to 1.81)	2.27
FFM (kg)	47.3 (41.1–61)	47 (41–60.8)	0.011	−0.84	(−1.05 to −0.064)	1.14	(0.94 to 1.34)	1.98
BMR (Kcal)	1494 (1306–1807)	1480 (1292–1801)	<0.001	−19.33	(−24.51 to −14.16)	30.82	(25.65 to 36.00)	50.15
Impedance (Ω) *	648 (564–752)	658 (561–752)	0.956	−35.21	(−42.53 to −27.90)	35.69	(28.38 to 43.01)	70.9
BIA_MF_ (*n* = 69 set 1; *n* = 70 set 2)								
TBW (L) †	33.3 (28.8–44.2)	33.1 (28.9–43.7)	<0.001	−0.61	(−0.79 to −0.43)	1.13	(0.95 to 1.31)	1.74
FM (kg) †	13.9 (11.3–18.2)	14.4 (11.2–18.3)	0.001	−1.05	(−1.32 to −0.77)	1.56	(1.28 to 1.83)	2.61
FFM (kg) †	45.5 (39.4–60.3)	45.4 (39.5–59.6)	<0.001	−0.81	(−1.06 to −0.56)	1.58	(1.33 to 1.83)	2.39
BMR (Kcal) †	1354 (1221–1672.5)	1350 (1223–1657)	<0.001	−17.76	(−23.24 to −12.27)	34.60	(29.11 to 40.08)	52.36
Impedance (Ω) *	246.2 (223.3–280)	246.4 (222.8–279.6)	0.150	−5.83	(−7.17 to −4.50)	6.87	(5.54 to 8.21)	12.7

TBW, total body water; FM, body fat mass; FFM, fat-free-mass; BMR, basal metabolic rate; * Impedance of whole body in BIA_SF_ and Impedance of the trunk in BIA_MF_. Set 1, data collected pre miction; Set 2, data collected after voiding; IQR, interquartile range;. †, Non-normal (Non-parametric analyses). *p*, *p*-value of mean difference by *t*-test (†). LOA, 95% límit of agreement. Range: Interval between lower and upper LOA.

**Table 4 sensors-24-07343-t004:** Inter-methods agreement in body composition variables with full bladder and after voiding.

				Bland Altman Analysis						
BIA_SF_ vs. BIA_MF_	ICC	IC 95%	$\bar{d}$ BIA_SF_-BIA_MF_ (SD)	IC (95%)	Lower LoA	IC (95%)	Upper LoA	IC (95%)	Range	Trend r
Set 1										
TBW (L)	0.98	(0.97 to 0.99)	1.17 (1.45)	(0.82 to 1.52)	−1.66	(−2.26 to −1.07)	4.01	(3.41 to 4.60)	5.67	−0.18 (ns)
FM (kg)	0.94	(0.91 to 0.96)	−1.48 (2.07)	(−1.98 to −0.98)	−5.54	(−6.39 to −4.69)	2.58	(1.73 to 3.43)	8.12	0.24 (ns)
FFM (kg)	0.98	(0.97 to 0.99)	1.57 (2.01)	(1.09 to 2.05)	−2.37	(−3.19 to −1.54)	5.51	(4.68 to 6.33)	7.88	−0.13 (ns)
BMR (Kcal)	0.97	(0.95 to 0.98)	125.19 (64.14)	(109.78 to 140.60)	−0.53	(−26.87 to 25.80)	250.91	(224.57 to 277.24)	251.44	
Set 2										
TBW (L)	0.98	(0.97 to 0.99)	1.35 (1.44)	(1.01 to 1.70)	−1.47	(−2.06 to −0.88)	4.18	(3.59 to 4.77)	5.65	−0.15 (ns)
FM (kg)	0.95	(0.92 to 0.97)	−1.72 (2.05)	(−2.21 to −1.23)	−5.74	(−6.58 to −4.90)	2.30	(1.46 to 3.14)	8.04	0.21 (ns)
FFM (kg)	0.98	(0.97 to 0.99)	1.85 (2.01)	(1.38 to 2.34)	−2.08	(−2.90 to −1.27)	5.80	(4.98 to 6.62)	7.88	−0.12 (ns)
BMR (Kcal)	0.97	(0.95 to 0.98)	129.83 (64.48)	(114.45 to 145.20)	3.45	(−22.83 to 29.72)	256.21	(229.93 to 282.49)	259.66	

Set 1: dates pre miction; Set 2: dates post voiding; TBW: total body water; FM: body fat mass; FFM: fat-free-mass; BMR: basal metabolic rate; $\bar{d}$, mean differences between set 1 and set 2; SD: standard deviations; ICC, Intraclass correlation coefficient; LoA: 95% limit of agreement; (ns), non significant.

## Data Availability

Data are available from the corresponding author upon reasonable request.

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
