# Peer review of "Influence of Bladder Filling on Parameters of Body Composition by Bioimpedance Electrical Analysis: Observational Study"

_sensors, 2024, doi:10.3390/s24227343_

Round 1

Reviewer 1 Report

Comments and Suggestions for Authors

Introduction:

In line 58 to 59 regarding the citation of reference 9 on the response to water-rich tissues, it would be appropriate to add a more recent reference on the subject. I can suggest Campa, F., Coratella, G., Cerullo, G. et al. High-standard predictive equations for estimating body composition using bioelectrical impedance analysis: a systematic review. J Transl Med 22, 515 (2024). Although more specific in the field of analysis on athletes, but the authors can certainly indicate one, in their opinion, more appropriate.

In closing the introduction, lines 65-74 it is necessary for the authors to extend the references, basing the closing of the chapter on two documets from the 90's is a bit too limiting, even if they are excellent scientific works: for example the NIH Consensus Statement, although a founding document for the field, is now superseded by many more recent studies that have expanded our understanding of the variations in body composition due to different frequencies, water conditions and other parameters.

M&M

Line 108, a specification that I don't think needs further clarification but, just to be safe... I take it for granted that the weight was measured as indicated "with the BIA" therefore taken before and after emptying the bladder, at the same time as the two analyses performed for each subject; it is appropriate that the authors specify this in the text for the safety of some distracted reader.

Discussion:

The discussion states that bladder filling affects BIA results, but no significant differences in impedance were found between full and empty bladder. This appears to contradict the initial hypothesis and subsequent statements highlighting significant differences in TBW, FFM, and FM values between these two conditions. This ambiguity would benefit from further explanation: if impedance does not change significantly, how can the observed differences in body composition results be explained? Moreover, the justification in the line "Despite not having observed differences in the impedance between both sets, both BIAS are sensitive enough to detect more TBW with full and empty bladder" could be expanded upon, possibly with some technical considerations regarding the instrument or analysis conditions.

The discussion mentions that the differences found between FM and FFM values with full and empty bladders are "not clinically significant" (line 275). However, this statement lacks a clear explanation of the criteria used to determine clinical significance. Providing more details on this, in light of the sensitivity of the instruments used, would clarify the importance of the observed differences.

It would also be helpful to add some commentary on the clinical utility of the two instruments in question, particularly given that the bladder filling factor appears to significantly influence the analysis. In the contexts where these instruments are typically used, are there additional considerations that should be taken into account?

(results-discussion) Since the sample consisted of 56.3% female subjects, it would be interesting to analyze potential gender differences in bioimpedance results related to bladder emptying. Women generally have a higher percentage of body fat than men, and this could have influenced the results for FM in the BIA analyses. Such an analysis would provide further insight into how bladder filling affects bioimpedance differently based on gender.

Another important aspect to explore could be the use of BIA devices that position the patient horizontally (supine) compared to those that leave the patient standing, as gravity might influence fluid distribution and, consequently, bioimpedance results. Additionally, devices that use disposable electrodes may have different conductivity than those with reusable electrodes, which could impact the accuracy of measurements, particularly in studies focusing on small variations like bladder content. Even a speculative discussion on how these differences might affect the measurements would contribute to a fuller understanding of the limitations and potential of these devices.

Conclusions

While the authors state that the two BIA methods are "not interchangeable" due to poor agreement, it would help to elaborate on how this impacts clinical or research applications. A mention of what specific applications might benefit from one method over the other would make this point more valuable to the reader.

Author Response

Dear Reviewer,

We would like to thank you for the constructive evaluation of our manuscript. We think that the comments are very useful, and they contribute to substantially improve the quality of our work. You will also find below our full rebuttal to all the comments made by both reviewers, so that a better-informed decision can be made regarding our study.

Reviewer 1:

Introduction:

In line 58 to 59 regarding the citation of reference 9 on the response to water-rich tissues, it would be appropriate to add a more recent reference on the subject. I can suggest Campa, F., Coratella, G., Cerullo, G. et al. High-standard predictive equations for estimating body composition using bioelectrical impedance analysis: a systematic review. J Transl Med 22, 515 (2024). Although more specific in the field of analysis on athletes, but the authors can certainly indicate one, in their opinion, more appropriate.

We agree with the reviewer and this point should be better supported. Thus, we add the following reference in line 61, as the reviewer suggested:

Campa, F, Coratella, G, Cerullo, G, Noriega, Z, Francisco, R, Charrier, D, Irurtia, A, Lukaski, H, Silva, AM, and Paoli A. High-Standard Predictive Equations for Estimating Body Composition Using Bioelectrical Impedance Analysis: A Systematic Review. J Anesth Transl Med 22, no. 1 (2024): 515.

In closing the introduction, lines 65-74 it is necessary for the authors to extend the references, basing the closing of the chapter on two documets from the 90's is a bit too limiting, even if they are excellent scientific works: for example the NIH Consensus Statement, although a founding document for the field, is now superseded by many more recent studies that have expanded our understanding of the variations in body composition due to different frequencies, water conditions and other parameters.

We agree with the reviewer's point. Thus, we add the following references (line 72):

Khalil, S. F., Mohktar, M. S., & Ibrahim, F. (2014). The theory and fundamentals of bioimpedance analysis in clinical status monitoring and diagnosis of diseases. Sensors (Basel, Switzerland), 14(6), 10895–10928. https://doi.org/10.3390/s140610895

Fosbøl, M. Ø., & Zerahn, B. (2015). Contemporary methods of body composition measurement. Clinical physiology and functional imaging, 35(2), 81–97. https://doi.org/10.1111/cpf.12152

Line 108, a specification that I don't think needs further clarification but, just to be safe... I take it for granted that the weight was measured as indicated "with the BIA" therefore taken before and after emptying the bladder, at the same time as the two analyses performed for each subject; it is appropriate that the authors specify this in the text for the safety of some distracted reader.

We appreciate the reviewer’s careful consideration of this point. We wrote the following sentence “Weight (kg) was measured with the BIA taken before and after bladder emptying, at the same time as the two analyses performed for each subject” to clearly indicate that weight was measured using the BIA device before and after bladder emptying, coinciding with the two analyses conducted for each subject.

Discussion:

The discussion states that bladder filling affects BIA results, but no significant differences in impedance were found between full and empty bladder. This appears to contradict the initial hypothesis and subsequent statements highlighting significant differences in TBW, FFM, and FM values between these two conditions. This ambiguity would benefit from further explanation: if impedance does not change significantly, how can the observed differences in body composition results be explained? Moreover, the justification in the line "Despite not having observed differences in the impedance between both sets, both BIAS are sensitive enough to detect more TBW with full and empty bladder" could be expanded upon, possibly with some technical considerations regarding the instrument or analysis conditions.

We recognize that the lack of significant differences in impedance between the full and empty bladder conditions may initially seem to contradict the observed changes in body composition parameters, specifically TBW, FFM, and FM.

To clarify, while impedance values remained stable, the BIA device is highly sensitive to minor variations in fluid distribution within the body. Even with unchanged impedance, the presence of additional bladder fluid likely influenced the algorithmic estimation of body composition components, leading to the detected differences in TBW, FFM, and FM. This phenomenon underscores the sensitivity of the device’s analytical algorithms, which interpret even subtle fluid shifts as significant changes in body composition outputs.

In response to the reviewer’s suggestion, we have expanded this section of the discussion (line 238-246) to include technical considerations of the BIA methodology and analysis conditions.

“However, the present study failed to find statistical differences in impedance variable (Ω) (Impedance of whole body in BIASF and Impedance of the trunk in BIAMF) were found between full and empty bladder. This finding may indicate that the findings are caused by the difference in weights rather than impedance. Regarding technical considerations of the BIA methodology and analysis conditions, we note that BIA devices use predictive equations that may factor in minor fluid changes without altering impedance values substantially. Additionally, these findings suggest that bladder filling could interact with the assumptions within these equations [7], impacting TBW estimations, even when impedance itself remains constant [29].”

This text is supported by:

  • Kyle, U. G., I. Bosaeus, A. D. De Lorenzo, P. Deurenberg, M. Elia, J. M. Gomez, B. L. Heitmann, L. Kent-Smith, J. C. Melchior, M. Pirlich, H. Scharfetter, A. M. Schols, C. Pichard, and Espen Working Group Composition of the. "Bioelectrical Impedance Analysis--Part I: Review of Principles and Methods." Clin Nutr 23, no. 5 (2004): 1226-43.
  • Moon, Jordan R, Sarah E Tobkin, Michael D Roberts, Vincent J Dalbo, Chad M Kerksick, Michael G Bemben, Joel T Cramer, and Jeffrey R Stout. Total Body Water Estimations in Healthy Men and Women Using Bioimpedance Spectroscopy: A Deuterium Oxide Comparison. Nutrition & metabolism 5 (2008): 1-6.

The discussion mentions that the differences found between FM and FFM values with full and empty bladders are "not clinically significant" (line 275). However, this statement lacks a clear explanation of the criteria used to determine clinical significance. Providing more details on this, in light of the sensitivity of the instruments used, would clarify the importance of the observed differences.

Thank you for this valuable observation. We agree that specifying the criteria for determining clinical significance is essential for clarity. In our study, we considered differences to be clinically significant if they exceeded the typical error margin reported for the BIA devices used, as well as thresholds established in previous research that indicate meaningful changes in body composition for clinical or performance outcomes. Given that the observed differences in all values fell within these established margins, we interpreted them as not clinically significant, suggesting that the bladder filling status does not substantially impact body composition estimates for practical use in most clinical contexts.

In response to your suggestion, we have clarified this in the abstract, discussion (line 284-290) and conclusion sections to provide readers with a better understanding of how we evaluated clinical relevance.

                Abstract:

                “Bladder filling does affect BIA measurements, not clinically meaningful”.

                Discussion:

“Although, in the present study, significant decreases were observed in all parameters (Table 3), previous research suggests that variations within approximately 1-2% of body composition metrics are typically within the measurement error of most BIA devices and are unlikely to be clinically meaningful [7, 36]. Since the observed differences values fell within these error margins, we interpreted them as not clinically significant, suggesting that bladder filling status does not substantially impact body composition estimates.”

References:

-Kyle, U. G., I. Bosaeus, A. D. De Lorenzo, P. Deurenberg, M. Elia, J. M. Gomez, B. L. Heitmann, L. Kent-Smith, J. C. Melchior, M. Pirlich, H. Scharfetter, A. M. Schols, C. Pichard, and Espen Working Group Composition of the. "Bioelectrical Impedance Analysis--Part I: Review of Principles and Methods." Clin Nutr 23, no. 5 (2004): 1226-43.

-Ling, C. H., de Craen, A. J., Slagboom, P. E., Gunn, D. A., Stokkel, M. P., Westendorp, R. G., & Maier, A. B. (2011). Accuracy of direct segmental multi-frequency bioimpedance analysis in the assessment of total body and segmental body composition in middle-aged adult population. Clinical nutrition30(5), 610-615.

Conclusion:

“The measurements of TBW, FM, FFM, BMR shown higher values with full bladder than with empty bladder, but these differences are not clinically relevant.”

It would also be helpful to add some commentary on the clinical utility of the two instruments in question, particularly given that the bladder filling factor appears to significantly influence the analysis. In the contexts where these instruments are typically used, are there additional considerations that should be taken into account?

Thank you for highlighting this point. We included this information in the introduction; however, we have include some particular use in clinical settings in line 319-324:

"In clinical practice, the choice between the BIAMF and BIASF devices should consider their distinct capabilities: while theBIAMF’s multi-frequency analysis and segmental assessment provide detailed insights into fluid compartments and localized body composition—beneficial in settings such as renal care, cardiology, or sports medicine—the  BIASF device offers a faster, single-frequency measurement suited for general health assessments where detailed fluid differentiation is less critical [12]".

Reference:

  • Yalin, S. F., S. Gulcicek, S. Avci, B. Erkalma Senates, M. R. Altiparmak, S. Trabulus, S. Alagoz, H. Yavuzer, A. Doventas, and N. Seyahi. "Single-Frequency and Multi-Frequency Bioimpedance Analysis: What Is the Difference?" Nephrology (Carlton) (2017).

(results-discussion) Since the sample consisted of 56.3% female subjects, it would be interesting to analyze potential gender differences in bioimpedance results related to bladder emptying. Women generally have a higher percentage of body fat than men, and this could have influenced the results for FM in the BIA analyses. Such an analysis would provide further insight into how bladder filling affects bioimpedance differently based on gender.

Thank you for this observation. “Although FFM and FM values typically differ between men and women—males generally having less fat mass and more fat-free mass across all age ranges [43]—we conducted our analysis with both genders combined. This approach was chosen because our primary objective was not to examine FFM or FM quantities by gender but rather to determine if these values changed after bladder emptying, regardless of gender.” This information has been included in line 325-329.

Reference:

-Kyle, U. G., Genton, L., Slosman, D. O., & Pichard, C. (2001). Fat-free and fat mass percentiles in 5225 healthy subjects aged 15 to 98 years. Nutrition17(7-8), 534-541.

Another important aspect to explore could be the use of BIA devices that position the patient horizontally (supine) compared to those that leave the patient standing, as gravity might influence fluid distribution and, consequently, bioimpedance results. Additionally, devices that use disposable electrodes may have different conductivity than those with reusable electrodes, which could impact the accuracy of measurements, particularly in studies focusing on small variations like bladder content. Even a speculative discussion on how these differences might affect the measurements would contribute to a fuller understanding of the limitations and potential of these devices.

The devices used in the present study are indeed limited to measuring participants in a standing position, which restricts our ability to explore how gravity may impact fluid distribution and bioimpedance results in a supine position. Additionally, our devices utilize fixed electrodes embedded in the device, so we cannot directly assess potential differences in conductivity between disposable and reusable electrodes.

However, we appreciate your suggestion to discuss these factors further, and we will include a speculative discussion in line 333-337 on how these positioning might influence measurement accuracy, especially in studies sensitive to small fluctuations, like bladder content analysis.

“Lastly, participant position (supine vs. standing) can impact BIA measurements due to gravitational shifts in fluid distribution. The devices used in the present study are limited to measuring participants in a standing position, which restricts our ability to explore how gravity may impact fluid distribution and bioimpedance results in a supine position.”

Conclusions

While the authors state that the two BIA methods are "not interchangeable" due to poor agreement, it would help to elaborate on how this impacts clinical or research applications. A mention of what specific applications might benefit from one method over the other would make this point more valuable to the reader.

Thank you for this suggestion. We agree that clarifying the implications of the poor agreement between the two BIA methods would strengthen the discussion. In response, we have added the following sentence to the conclusion:

"Both BIA methods demonstrated poor agreement, indicating they are not interchangeable. In clinical and research applications the BIASF provides consistently higher values for TBW and FFM, and consequently lower values for FM than the BIAMF”

Reviewer 2 Report

Comments and Suggestions for Authors

The study aims to evaluate the impact of bladder filling on bioimpedance measurements, including raw impedance and processed body composition values. This is a topic of interest where new data is of value. The study design is well thought through and gives significant consideration to controlling the conditions of measurement, ensuring robust data, which is to be commended. I feel that some improvement in the interpretation of the results and how the findings are presented could improve the manuscript further.

1)      Abstract: Suggest including the lack of difference in measured impedance values between the two states, demonstrating that the findings are caused by the difference in weights rather than impedance.

2)      Abstract: My interpretation of the findings would be that consistency of measurement conditions that can influence weight (e.g. GI content, bladder content, clothing) is extremely important. To maximise accuracy, conditions should best match the conditions that the equations for the specific device were developed in.

3)      Methods (Statistical analysis): Can you justify the approaches to analyse the data? Firstly, if you are using a t-test to compare devices, why are you not using a paired t-test? Also, why use different approaches for analysing agreement between states (full and empty bladder) and for analysing agreement between devices? Your statistical analysis says you will compare states using ICCs and Bland Altman but it looks like the primary results are based on a p value from t-test/non parametric test. I think it need to be very clear what the basis of the primary outcome is.

4)      Section 4.1. This should focus on the fact that the differences observed are directly linked to differences in weight and nothing to do with impedance measured.

Author Response

Dear Reviewer,

We would like to thank you for the constructive evaluation of our manuscript. We think that the comments are very useful, and they contribute to substantially improve the quality of our work. You will also find below our full rebuttal to all the comments made by both reviewers, so that a better-informed decision can be made regarding our study.

Reviewer 2:

The study aims to evaluate the impact of bladder filling on bioimpedance measurements, including raw impedance and processed body composition values. This is a topic of interest where new data is of value. The study design is well thought through and gives significant consideration to controlling the conditions of measurement, ensuring robust data, which is to be commended. I feel that some improvement in the interpretation of the results and how the findings are presented could improve the manuscript further.

1)      Abstract: Suggest including the lack of difference in measured impedance values between the two states, demonstrating that the findings are caused by the difference in weights rather than impedance.

Thank you for your help, we have added the following sentence in the abstract (line 26-27:

“TBW, FM, and BMR values were significantly reduced after bladder voiding possibly caused by the difference in weights rather than impedance (p>0.05)”.

2)      Abstract: My interpretation of the findings would be that consistency of measurement conditions that can influence weight (e.g. GI content, bladder content, clothing) is extremely important. To maximise accuracy, conditions should best match the conditions that the equations for the specific device were developed in.

We have included measurement conditions in the abstract (line 22-23). Thank you for your suggestion.

“Participants in underwear, followed an overnight fast. They were instructed to abstain from vigorous physical activity and alcohol for at least 24 hours prior to the session.”

3)      Methods (Statistical analysis): Can you justify the approaches to analyse the data? Firstly, if you are using a t-test to compare devices, why are you not using a paired t-test? Also, why use different approaches for analysing agreement between states (full and empty bladder) and for analysing agreement between devices? Your statistical analysis says you will compare states using ICCs and Bland Altman but it looks like the primary results are based on a p value from t-test/non parametric test. I think it need to be very clear what the basis of the primary outcome is.

We used independent t-tests for preliminary comparisons because, even though it is the same subject, the measurement from one device does not influence the measurement from the other.

Regarding the choice of methods for agreement analysis, we understand that Bland-Altman plots are typically interpreted informally, focusing on clinical interpretation rather than further statistical testing. In our analysis, we use it as complementary analysis emphasized the clinical significance of the bias. The Bland-Altman plots revealed considerable discrepancies between the devices, especially in FM and FFM, with discrepancies for FFM between devices, even in the same bladder condition. These differences are clinically significant, reinforcing that the devices are not interchangeable. For clarity, we have now emphasized that these clinically meaningful differences are our primary outcome, and the t-test findings serve as supporting data in line 149-150.

“Bland-Altman analysis [27] was used as a complementary analysis for clinically meaningful differences”.

Thank you again for highlighting these points, which helped refine the presentation and interpretation of our results.

4)      Section 4.1. This should focus on the fact that the differences observed are directly linked to differences in weight and nothing to do with impedance measured.

We agree with the reviewer and this point should be better clarify, in this paragraph we were talking about the variable of bioimpedance, not about the method, but we did not specify properly. Thus, as reviewer said, that lines seem to be contradictories. For a better understanding we add in the discussion section the following text (line 238-241):

“However, the present study failed to find statistical differences in impedance variable (Ω) (Impedance of whole body in BIASF and Impedance of the trunk in BIAMF) were found between full and empty bladder. This finding may indicate that the findings are caused by the difference in weights rather than impedance”.

Round 2

Reviewer 2 Report

Comments and Suggestions for Authors

Thank you for addressing my comments. Two suggestions remain:

1) I think the abstract could be more explicit about the findings, as it is written it could be misinterpreted (the measured impedance did not change at all so I don't think it is accurate to say "possibly" in this context. I would change the section:

"The findings suggest that bladder filling significantly influences BIA results. TBW, FM, and BMR values were significantly reduced after bladder voiding, possibly caused by the difference in weights rather than impedance (p>0.05)."

to something like:

"The findings suggest that bladder filling has no impact on measured impedance but the change in weight influenced derived BIA results. TBW, FM, and BMR values were significantly reduced after bladder voiding (p>0.05)."

2) You have paired data (measurements by two devices on the same individual), so a paired t-test is more appropriate than an unpaired t-test which would be used for two independent populations.

Author Response

Dear Reviewer,

We would like to thank you for the constructive evaluation of our manuscript.

Reviewer R2:

Thank you for addressing my comments. Two suggestions remain:

1) I think the abstract could be more explicit about the findings, as it is written it could be misinterpreted (the measured impedance did not change at all so I don't think it is accurate to say "possibly" in this context. I would change the section:

"The findings suggest that bladder filling significantly influences BIA results. TBW, FM, and BMR values were significantly reduced after bladder voiding, possibly caused by the difference in weights rather than impedance (p>0.05)."

to something like:

"The findings suggest that bladder filling has no impact on measured impedance but the change in weight influenced derived BIA results. TBW, FM, and BMR values were significantly reduced after bladder voiding (p>0.05)."

Thank you for your suggestion. We have replaced this sentence with:

"The findings suggest that bladder filling does not affect measured impedance; however, changes in weight following bladder voiding influenced derived BIA results. Specifically, TBW, FM, and BMR values significantly reduced after voiding (p < 0.05)."

2) You have paired data (measurements by two devices on the same individual), so a paired t-test is more appropriate than an unpaired t-test which would be used for two independent populations.

Thank you for your comment. Although the measurements were obtained from the same individuals, we directly compared the data collected from the two devices. For this reason, we believe a paired sample analysis may not be appropriate, as we are assessing potential differences attributable to the devices themselves rather than repeated measures on identical samples.

However, if you believe a paired t-test would still add value to the analysis, we are open to including it despite our reservations.
